

# invgamma: the inverse gamma distribution in R

David Kahle and James Stamey

Department of Statistical Science, Baylor University, Waco, Texas, United States

## ABSTRACT

invgamma is a popular low dependency R package that implements the probability density function (PDF), cumulative distribution function (CDF), quantile function (QF) and random number generator (RNG) functions for the inverse gamma, inverse chi-squared, and inverse exponential distributions, which are missing from base R. The functions follow the standard R syntax and are efficient, leveraging the corresponding functions for the gamma distribution currently in R through straightforward mathematical relationships between the distributions. It is distributed through the Comprehensive R Archive Network (CRAN, https://cran.r-project.org) and GitHub (https://github.com/dkahle/invgamma), where it is version controlled.

## INTRODUCTION

R is the programming lingua franca of academic statisticians and one of the most popular statistics and data science programming languages (*R Core Team, 2016*; *Muenchen, 2023*). In its base distribution it contains core functionality for all sorts of operations essential to a proper platform for statistical computing: reading and writing files, data manipulation, graphical capabilities, scientific computing data structures and algorithms (*e.g.*, arithmetic, matrix algebra, sorting, and special mathematical functions), modeling capabilities (*e.g.*, regression), and much more. R also has a thriving developer base creating contributed packages, which number in the tens of thousands at the time of this writing, hosted on R's primary package repository, the Comprehensive R Archive Network (CRAN, https://cran-r-project.org).

One of the key components of R is a simple and efficient code base for working with probability distributions. R has dozens of built-in functions related to common probability distributions such as probability density functions (PDFs), cumulative distribution functions (CDFs), quantile functions (QFs), and random number generators (RNGs), typically implemented as compiled C executables. For a given family of probability distributions, these four functions follow a memorable naming pattern: `[dpqr]<dist>()`, where the letters d, p, q, and r signify the PDF, CDF, QF, and RNG of the distribution (respectively), and `<dist>` is the short-hand form of the family name that R uses. Thus, `dnorm()` is the PDF of the normal distribution; `qbinom()` is the quantile function of the binomial distribution; and `rt()` is the RNG for the *t* distribution. We refer to these as the dpqr functions, and for brevity when referring to the collection of four for a particular family write `*<dist>()`, *e.g.*, `*gamma()`. The specific parameterizations of these families are communicated in their corresponding

Corresponding author
David Kahle,
david_kahle@baylor.edu

documentation accessible with ?, *e.g.*, ?dnorm. These functions are all provided by the **stats** package, one of R's base packages that is loaded by default when an R session is initialized.

Conspicuously missing from the list, however, is the inverse gamma distribution and its related subfamilies: the inverse $\chi^2$ and inverse exponential distributions. In this short article we discuss invgamma, a low-dependency, efficient, and low-maintenance package available on CRAN that has provided the functions to fill this void for the past decade, with downloads totaling well over three hundred thousand (*Csárdi, 2019*). Although previously only a software implementation, we, the authors of invgamma, write this article now to reflect on the software, present the core mathematics behind it (some of which is sparsely documented elsewhere), and clarify a few design choices.

## GETTING INVGAMMA

There are two ways to obtain invgamma. First, one can obtain it from CRAN using R's built-in packaging tool install.packages():

```
install.packages ("invgamma")
```

Alternatively, one can install the most recent stable development version of the package using **remotes** (*Csárdi et al., 2024*):

```
remotes:: install_github ("dkahle/invgamma")
```

The package is then loaded and attached to the current R session using the standard mechanism:

```
library ("invgamma")
```

## MATHEMATICAL UNDERPINNINGS

### The gamma distribution and its implementation in R

This article is interested in the inverse gamma distribution, which is defined in reference to the gamma distribution. The gamma distribution is so called because it relies fundamentally on the gamma function, a special function defined for all complex $x$ with real part greater than zero as the definite integral

$$\Gamma(x) = \int_0^\infty t^{x-1}e^{-t}\,dt, \qquad \text{Re}(x) > 0. \tag{1}$$

It is well known that this function cannot be expressed in terms of elementary functions but exhibits many very special properties.

Where the relevant quantities are defined, one important property is the recurrence identity $\Gamma(x) = (x-1)\Gamma(x-1)$. This property enables "shifting left" in the sense of allowing the computation of $\Gamma$ at $x$ by computing it its value 1 to the left, *i.e.*, $x-1$. It also enables "shifting right," computing $\Gamma$ at $x$ by computing it 1 to the right, *i.e.*, $x+1$. Together, these are

$$(x-1)\Gamma(x-1) = \Gamma(x) = \frac{1}{x}\Gamma(x+1). \tag{2}$$

When applied to positive integers $n = x$, the recurrence identity reveals that the Gamma function is a generalization of the factorial function communicated in postfix notation, $n! := \Gamma(n+1) = n\Gamma(n) = n(n-1)\Gamma(n-1) = n(n-1)(n-2)\cdots(2)\Gamma(1) = n(n-1)(n-2)\cdots(2)(1) = \prod_{i=1}^{n} i$.

Another identity, valid for non-integer $x$, is Euler's reflection formula

$$\Gamma(x) = \frac{\pi}{\sin(\pi x)\Gamma(1-x)}. \tag{3}$$

This formula is particularly important as it allows the definition (more, an analytic continuation) of the gamma function over whole complex plane minus zero and the negative integers. For example, when presented with a negative number[1] the reflection formula can be applied to recast the problem as one over the positive numbers. For example, $\Gamma(-3.5) = \frac{\pi}{\sin(\pi(-3.5))\Gamma(1-(-3.5))} = \frac{\pi}{\sin(\pi(-3.5))\Gamma(4.5)}$.

Computing the gamma function is a well-studied problem in numerical analysis (*Schmelzer & Trefethen, 2007*; *Lange, 2010*). The classic reference for numerical approximations is *Abramowitz & Stegun (1964)*; a more modern reference is National Institute of Standards and Technology's (NIST's) Digital Library of Mathematical Functions (DLMF) (*NIST, 2025*).[2] In R, $\Gamma(x)$ is available *via* the function `gamma()`, which calls a compiled C function called `gammafn()`, itself a translation of an earlier Fortran routine attributed in the source code to W. Fullerton.[3] `gammafn()` accepts any number represented as a `numeric`, *i.e.*, a floating point or integer, and evaluates the function using a multi-regime design:

1. If $x \leq 0$ and $x$ is integer; return `NaN` and issue a warning.
2. If $1 \leq x \leq 50$ and $x$ is an integer, brute-force the product $(x-1)!$ by looping.
3. If $x < 0$ is non-integer, apply the reflection formula and compute $\Gamma(1-x)$ as below.
4. If $|x| \leq 10$, shift $x$ left or right *via* the recurrence relation until it lies in $[1, 2)$ and then evaluate $\Gamma(1+y) = \int_0^\infty t^y e^{-t}\, dt$ with a 42-term Chebyshev approximant whose coefficients are hard-coded into the source code, where $y$ is the factional part after reduction. For example, $\Gamma(3.2) = 2.2\Gamma(2.2) = (2.2)(1.2)\Gamma(1.2)$ so that $y = 0.2$ or $\Gamma(-2.2) = \frac{\pi}{\sin(-2.2\pi)\Gamma(1-(-2.2))} = \frac{\pi}{\sin(-2.2\pi)\Gamma(3.2)}$.
5. If $|x| > 10$, if more than about 171 return `Inf` or less than about $-171$ return 0, otherwise use Stirling's approximation $\Gamma(x) = \exp\{\frac{1}{2}\log 2\pi + (x-1/2)\log x - x + c\}$, where $c$ is a correction term with its own hard-coded Chebyshev approximant.

This design is illustrative of the general character of evaluating special functions in R. Functions directly related to the gamma function, *e.g.*, its natural logarithm implemented in R as `lgamma()`, often use C-level manipulations of these.[4] So, for example, `lgamma()` calls the C-function `lgammafn()` which either calls `gammafn()` and computes its log or uses a reflection formula trick, depending on the input value. In other words, the C-level implementation `gammafn()` is central to not just computing $\Gamma(x)$, but related functions as well.

[1] Negative real part, that is.

[2] The DLMF gamma function section can be found at https://dlmf.nist.gov/5.

[3] The source code is viewable at https://svn.r-project.org/R/tags/R-4-5-0/src/nmath/gamma.c.

[4] The source code is viewable at https://svn.r-project.org/R/tags/R-4-5-0/src/nmath/lgamma.c.

The PDF of the gamma distribution is one of these, albeit through a somewhat circuitous route. Suppose a random variable $X$ follows a gamma distribution with parameters $\alpha$ and $\lambda$, written $X \sim \text{Gamma}(\alpha, \lambda)$. By definition, this means that its PDF $f_X(x|\alpha, \lambda)$ is

$$f_X(x|\alpha, \lambda) = \frac{\lambda^\alpha}{\Gamma(\alpha)} x^{\alpha-1} e^{-\lambda x}, \qquad x, \alpha, \lambda > 0. \tag{4}$$

This parameterizaton $(\alpha, \lambda)$ is the *rate* parameterization of the family, whose mean is $\mu_X = \alpha/\lambda$ and variance is $\sigma_X^2 = \frac{\alpha}{\lambda^2}$. The Greek character $\lambda$ is commonly used to emphasize this parameterization, especially in relation to the Poisson distribution. The rate parameterization is often contrasted with the *scale* parameterization $(\alpha, \beta)$, which uses the Greek character $\beta$ defined $\beta = \frac{1}{\lambda}$, resulting in the PDF $f_X(x|\alpha, \beta) = \frac{1}{\Gamma(\alpha)\beta^\alpha} x^{\alpha-1} e^{-x/\beta}$, the mean $\mu_X = \alpha\beta$, and the variance $\sigma_X^2 = \alpha\beta^2$ (*Casella & Berger, 2002*). These parameters index the same gamma family of probability distributions and in this sense are equivalent, but they are not identical. The convention matters when actual formulas are desired for computing, *e.g.*, which convention is signified by $f_X(x|2, 3)$.

The function $f_X(x|\alpha, \lambda)$ is implemented in R as `dgamma()`, which like other `dpqr` functions is provided by **stats**. In `dgamma()`, the $\alpha$ and $\lambda$ parameters go by the names `shape` and `rate`. The formal arguments of `dgamma()` begin with `x`, `shape`, and `rate`, so that R's standard argument matching rules interpret un-named arguments as the $(\alpha, \lambda)$ parameterization, *i.e.*, `dgamma(x, 2, 3)` is interpreted as $f_X(x|\alpha = 2, \lambda = 3)$ (*Wickham, 2014*). However, the *scale* parameterization $\text{Gamma}(\alpha, \beta)$ is also available *via* `dgamma()`'s `scale` argument, which can be used by explicitly referring to it, *e.g.*, `dgamma(x, 2, scale = 1/3)`. Indeed, all the `dpqr` functions of the gamma distribution accept these parameter arguments.

Internally, `dgamma()` calls a C-level function by the same name; however, this function does not directly call the C function `gammafn()` or its log variant.[5] Instead, it calls an implementation of the probability mass function (PMF) of a Poisson random variable $f_Y(y|\lambda) = \frac{e^{-\lambda}\lambda^y}{y!} = \frac{e^{-\lambda}\lambda^y}{\Gamma(y+1)}$ *via* a trick that massages a scale-parameterized gamma PDF $f_X(x|\alpha, \beta)$ into a Poisson PMF:

$$f_X(x|\alpha, \beta) = \frac{1}{\Gamma(\alpha)\beta^\alpha} x^{\alpha-1} e^{-x/\beta} = \frac{1}{\beta} \frac{e^{-x/\beta}(x/\beta)^{\alpha-1}}{\Gamma(\alpha)} = \frac{1}{\beta} f_Y(\alpha - 1|\lambda = x/\beta) = \frac{P[Y = \alpha - 1]}{\beta}, \tag{5}$$

where $Y \sim \text{Pois}(\lambda = x/\beta)$. The function that implements the Poisson PMF at the C-level, called `dpois_raw()`, does call `lgammafn()` (among other logic), further illustrating the interdependence of these functions. An illustration of this equivalence can be seen below.

```
x <- 2.7; alpha <- 2; beta <- 3

dgamma (x, shape = alpha, scale = beta)

## [1] 0.1219709

dpois (alpha-1 , lambda = x/beta)/beta

## [1] 0.1219709
```

[5] The source code can be viewed at https://svn.r-project.org/R/tags/R-4-5-0/src/nmath/dgamma.c.

The cumulative distribution function of the gamma distribution is $F_X(x|\alpha, \lambda) = \int_0^x f_X(t|\alpha, \lambda)\, dt\ 1[x \geq 0]$, a scaled incomplete gamma function, where $1[x \geq 0]$ is the indicator function assuming the value 1 when $x \geq 0$ and 0 otherwise. It is implemented in R as `pgamma()`.[6] Like the gamma function itself, the actual computations are performed *via* compiled C code that uses a multi-regime design, which is here a bit more complex but uses the same core functions (*e.g.*, `dpois_raw()`) and approximation theory concepts. The computations are sometimes done first on a log scale, which allow for better numerical properties when the output values' magnitudes are very large or very small. These can be returned by `dgamma()` and `pgamma()` as well by specifying `log = TRUE` and `log.p = TRUE`, which implement log $f_X(x|\alpha, \lambda)$ and log $F_X(x|\alpha, \lambda)$, respectively.

The quantile function of the gamma distribution, the inverse-function of its CDF, is denoted $F_X^{-1}(x|\alpha, \lambda)$. If the CDF solves the problem "Given $x$ in $P[X \leq x] = p$, compute $p$.", the QF solves the problem "Given $p$ in $P[X \leq x] = p$, compute $x$." It is implemented in R as `qgamma()` and, like the other functions, is computed *via* compiled C code.[7] Like `gamma()` and `pgamma()`, the evaluation uses results from approximation theory, in this case using a reformulation of the gamma problem into a $\chi^2$ one to provide an initial guess followed by 7-term Taylor series correction using pre-computed coefficients (*Best & Roberts, 1975*). Newton steps are then performed as-needed to obtain full double-precision accuracy.

Random number generation from the gamma is implemented in `rgamma()` using a modified rejection sampler implemented in C (*Ahrens & Dieter, 1982, 1974*).[8] The construction is a complicated composite of a few simple regimes, the first of which applies when the shape $\alpha < 1$ and the second, more complex, when $\alpha \geq 1$. The specific components generally following standard rejection and envelope rejection sampler strategies (*Robert, Casella & Casella, 1999*).

With these tools in hand, we are now able to discuss the relevant functions for the inverse gamma distribution used in invgamma.

### The inverse gamma distribution and its connection to the gamma

The *inverse* gamma distribution is the distribution of a random variable $Y = X^{-1}$ whose inverse is a Gamma$(\alpha, \lambda)$ random variable, succinctly written $Y \sim$ Inv-Gamma$(\alpha, \lambda)$. Derived below, the PDF of the inverse gamma distribution is

$$f_Y(x|\alpha, \lambda) = \frac{\lambda^\alpha}{\Gamma(\alpha)} x^{-(\alpha+1)} e^{-\lambda/x} \qquad x, \alpha, \lambda > 0. \tag{6}$$

One of the standard families of probability distributions, the inverse gamma distribution showcases strongly in Bayesian statistics, where it is the conjugate prior for the variance of a normal data model with unknown mean and variance (*Christensen et al., 2011*; *Lunn et al., 2012*). More broadly, it is a common choice for a continuous probability distribution on the non-negative real numbers.

One of the nice features of the invgamma implementation is that it leverages the `dgamma()`, `pgamma()`, `qgamma()`, and `rgamma()` suite of functions shipped with R and maintained by the R core team. It does so *via* simple mathematical relationships between

[6] Viewable at https://svn.r-project.org/R/tags/R-4-5-0/src/nmath/pgamma.c.

[7] Viewable at https://svn.r-project.org/R/tags/R-4-5-0/src/nmath/qgamma.c.

[8] Viewable at https://svn.r-project.org/R/tags/R-4-5-0/src/nmath/rgamma.c.

the gamma and inverse gamma distributions. In this section we briefly describe these relationships. Throughout, it is assumed that $X \sim \text{Gamma}(\alpha, \lambda)$ and $Y = \frac{1}{X}$. We begin with the CDF.

Since $Y = X^{-1}$, the CDF of the inverse gamma can be easily described in terms of the CDF of the gamma distribution. In particular, if $F_Y(x)$ is the CDF of the inverse gamma distribution and $F_X(x)$ is the CDF of the corresponding gamma distribution, for $x > 0$ the complement rule and continuity require

$$F_Y(x) = P[Y \leq x] = P[X^{-1} \leq x] = P[1/x \leq X] = 1 - P[X < 1/x] = 1 - F_X(1/x). \quad (7)$$

The `pinvgamma()` function is implemented by wrapping `pgamma()` in this way, leveraging the `lower.tail` and `log.p` arguments of the `pgamma()` function to allow users to access those same arguments, which is consistent with their expectations set by the rest of the R ecosystem. Similar conventions analogous to the `*gamma()` functions are adopted when users specify special values, *e.g.*, `dinvgamma(0, 3, 4)` results in 0 (not `NaN` which is the result of division by zero), as does `dinvgamma(Inf, 3, 4)`.

A standard result in mathematical statistics known as the transformation theorem states that if $g$ is a differentiable, monotonic function and $Y = g(X)$, then the PDF of $Y$ satisfies $f_Y(x) = f_X(g^{-1}(x))|\frac{d}{dx}g^{-1}(x)|$ (*Casella & Berger, 2002*). For the inverse gamma distribution $g(x) = \frac{1}{x}$ so that $g^{-1}(y) = \frac{1}{y}$ and

$$\log f_Y(x) = \log\left(f_X(g^{-1}(x))\left|\frac{d}{dx}g^{-1}(x)\right|\right) = \log f_X(1/x) - 2\log x.$$

This allows us to construct `dinvgamma()` by computing `dgamma()` on the log scale (`log = TRUE`) and exponentiating as needed. The `log` argument is implemented in `dinvgamma()` as it is often used in likelihood computations; it is also standard in R for `d*()` functions (PDFs) to have this argument.

The quantile function is the inverse-function of the CDF. Using the variable-swapping method of determining function inverses, we start with

$$p = P[Y \leq x] = 1 - F_X(1/x), \quad (8)$$

swap variables, and solve to determine $x = F_X^{-1}(1-p)^{-1}$. Thus, `qinvgamma()` is easily expressible in terms of `qgamma()`: $F_Y^{-1}(p) = \frac{1}{F_X^{-1}(1-p)}$. Like `dgamma()`'s `log` and `pgamma()`'s `log.p` arguments, `qgamma()` can be used on the log scale; it admits the `log.p` argument. Naturally (although not obviously), when set to `TRUE` the input argument p to `qgamma()` is assumed to be on the log scale, *i.e.*, $\log p$.

As previously described, in R CDF and quantile functions are often given two other arguments: `lower.tail` and `log.p`. The first of these is used to compute the survival function $1 - F_Y(x)$ and the other to provide the probability on the log scale. Both of these are included in both the CDF and quantile functions by simply passing the arguments to `[pq]gamma()`.

Generating inverse gamma variates is the easiest of the four functions. Since by definition inverse gamma variates are simply the inverses of gamma variates, the

`rinvgamma()` function just takes `rgamma()` variates and inverts them using R's vectorized arithmetic operators, which are implemented at the C level.

Altogether, these produce the following four functions and signatures:

```
dinvgamma(x, shape, rate = 1, scale = 1/rate, log = FALSE)

pinvgamma(q, shape, rate = 1, scale = 1/rate, lower.tail = TRUE,
          log.p = FALSE)

qinvgamma(p, shape, rate = 1, scale = 1/rate, lower.tail = TRUE,
          log.p = FALSE)

rinvgamma(n, shape, rate = 1, scale = 1/rate)
```

These are identical to those of their `*gamma()` counterparts, making for a nice, intuitive experience for R users.

## ILLUSTRATIVE EXAMPLES

In this section we briefly show how the functions can be used and demonstrate their correctness. We begin by visualizing the $Y \sim \text{Inv-Gamma}(7, 10)$ distribution (*Wickham, 2009*; *Wickham et al., 2019*). The corresponding PDF is included in Fig. 1.

```r
# load helper packages, installing where necessary
install_if_needed <- function(pkg){
  if (!requireNamespace (pkg)) install.packages (pkg)
}
install_if_needed ("tidyverse"); library ("tidyverse")
install_if_needed ("patchwork"); library ("patchwork")
library ("scales", warn.conflicts = FALSE)
theme_set(theme_minimal())
theme_update (panel.grid.minor = element_blank())
# plot the inverse gamma PDF
ggplot() +
  stat_function(
    fun = dinvgamma, n = 251, xlim = c(0, 6), alpha = .65,
    args = list (shape = 7, rate = 10), geom = "polygon"
  ) + theme (axis.title = element_blank())
```

This also illustrates that the function `dinvgamma()` is vectorized across its first argument, as R users expect. This is a really nice feature for likelihood estimation, where the likelihood function is evaluated at several points and multiplied. Typically computed

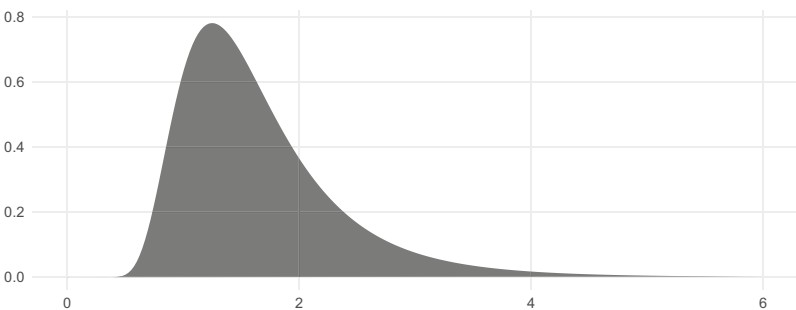

**Figure 1** **The PDF of the Inv-Gamma (7, 10) distribution can be visualized with `dinvgamma()`.**

on the log-scale, we can pass in the argument `log = TRUE` and apply `sum()` to affect a similar, but more numerically useful, result.

The consistency of `pinvgamma()` with `dinvgamma()` can be checked with the `integrate()` function, which applies numerical quadrature *via* adaptive Gauss–Kronrod quadrature. Here, we compute $F_Y(2) = P[Y \leq 2]$ using both `integrate(dinvgamma())` and `pgamma()`:

```
f <- function(x) dinvgamma (x, 7, 10)

q <- 2

integrate (f, 0, q)
## 0.7621835 with absolute error < 7.3e-05
(p <- pinvgamma (q, 7, 10))
## [1] 0.7621835
```

Since these values are equivalent, `dinvgamma()` and `pinvgamma()` are satisfying the expected relationship consistently. Checking `qinvgamma()`'s consistence is a simple step from here: we simply take the inverse of $P[Y \leq 2]$, expecting 2.

```
qinvgamma (p, 7, 10) # = q
## [1] 2
```

The above illustrates that `dinvgamma()`, `pinvgamma()`, and `qinvgamma()` are consistent in the sense that they satisfy the probabilistic relationships demanded among their respective functions: they refer to the same probability distribution. To demonstrate that they are sampling from the correct distribution, we compute the same quantity as before, $F_Y(2)$, using Monte Carlo simulation *via* `rinvgamma()`. Since `rinvgamma()` simply uses `1/rgamma()`, it samples from the correct distribution by definition; its numerics are addressed shortly.

```
set.seed (1)

n <- 1e5

draws <- rinvgamma (n, 7, 10)

mean (draws <= q)

## [1] 0.76209
```

Since 0.76209 is within the Monte Carlo error of 0.7621835 (the true value computed previously) when using 100,000 draws, it is clear that the implementations are correct. As more complex demonstrations, in Fig. 2 we superimpose a kernel density estimate of the 100,000 sampled values with the density given by `dinvgamma()` and present a QQ-plot of the draws, both of which strongly confirm the correctness of the sampler (*Pedersen, 2024*). Figure 2 is generated with the following code:

```
df <- data.frame ("x" = draws)

p_hist <- ggplot (df, aes(x = x)) +

  geom_histogram (aes(y = after_stat (density)), bins = 250) +

  geom_function(fun = f, color = "red", n = 251 ) +

  coord_cartesian (xlim = c (0, 6))

qq_df <- data.frame(

  "theoretical_quantiles" = ppoints (n) |> qinvgamma (7, 10),

  "observed_quantiles" = sort (draws)

)

p_qq <- ggplot (qq_df, aes (theoretical_quantiles, observed_quantiles)) +

  geom_point () +

  geom_abline(slope = 1, intercept = 0, color = "red") +

  labs ("x" = "Theoretical Quantiles", "y" = "Observed Quantiles")

p_hist + p_qq
```

As a final note, like their *gamma() counterparts, *invgamma() functions accept both the `rate` (default) and `scale` parameters. Their consistency can be seen in the following example with `dinvgamma()`:

```
dinvgamma (.75, shape = 7, rate = 10)

## [1] 0.2246903

dinvgamma (.75, shape = 7, scale = 1/10)

## [1] 0.2246903
```

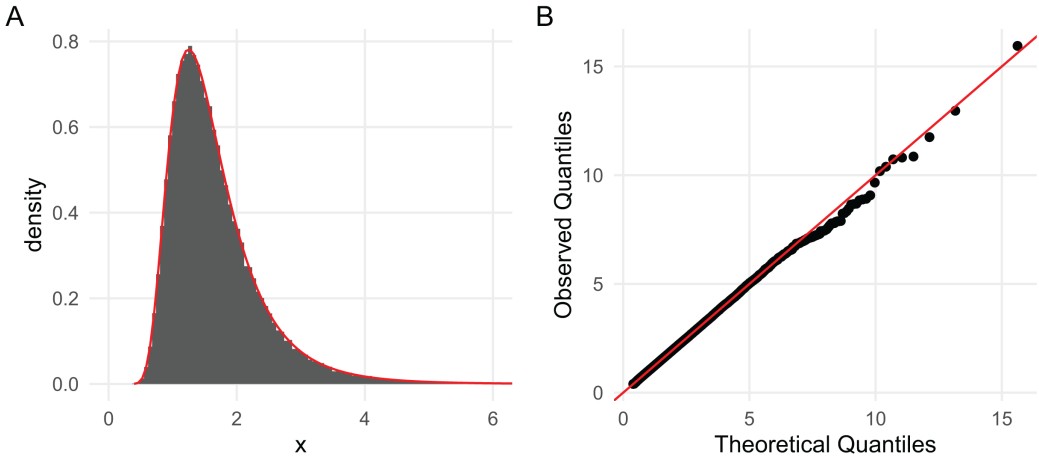

**Figure 2** (A) A histogram based on 100,000 draws generated with `rinvgamma()` superimposed with the inverse gamma density provided by `dinvgamma()` (red); and (B) a QQ plot of the estimated quantiles (order statistics) *vs.* theoretical quantiles computed with `qinvgamma()`. Both strongly suggest the RNG is correct.

## THE INVERSE $\chi^2$ AND INVERSE EXPONENTIAL DISTRIBUTIONS

The mathematical arguments made in the previous section work for any continuous distribution with a smooth density defined on the positive real numbers. Particularly notable among this class of distributions are two commonly used families subsumed by the gamma distribution: the $\chi^2$ (chi-squared) distribution and the exponential distribution. Since these two are so closely related to the gamma distribution, we have implemented the entire `dpqr` line of functions for the inverses of both of these distributions in **invgamma** as well.

The chi-squared distribution with $v$ degrees of freedom, denoted $\chi^2_v$, is equal to the $\text{Gamma}\left(\alpha = \frac{v}{2}, \lambda = \frac{1}{2}\right)$ distribution; it is the distribution of the sum of $v$ independent squared standard normal variates. In **invgamma**, the $*\text{invchisq()}$ functions for the inverse $\chi^2$ distribution could be implemented through the $*\text{invgamma()}$ functions, but we have chosen to model them off of the $*\text{chisq()}$ functions in the same way that the $*\text{invgamma()}$ functions are implemented using the $*\text{gamma()}$ functions, for two reasons. First, the $*\text{chisq()}$ functions allow for non-central $\chi^2$ distributions, which are not special cases of the inverse gamma but nevertheless can be obtained by logic similar to that of the previous section. Second, using the $*\text{chisq()}$ functions insures that the $*\text{invchisq()}$ functions are synchronized with base R's implementation of the $\chi^2$ distribution, which is more natural than syncing them with the $*\text{gamma()}$ implementations.

The exponential distribution, like the gamma distribution, also manifests in a rate and a scale parameterization; both are obtained by setting the gamma distribution's shape parameter $\alpha$ to 1. However, only the rate parameterization is implemented in R's $*\text{exp()}$ functions. invgamma stays true to this convention in its $*\text{invexp()}$ functions, which have a `rate` argument defaulted to 1. And, like the $*\text{invchisq()}$ functions, the $*\text{invexp()}$

functions wrap the $*$exp() functions, not the $*$invgamma() functions. As in the inverse $\chi^2$ case, this allows the functions to naturally sync with the $*$exp() functions.

## A NOTE ON PARAMETERIZATION AND ARGUMENT NAMES

The point of invgamma is to provide a simple, low-maintenance implementation of the d, p, q, and r functions of the inverse gamma, chi-squared, and exponential distributions. To do this, it uses the transformation theorem along with the implementations of the corresponding functions for the gamma distribution, which are already implemented in **stats** as dgamma() and so on. The argument names were intended to reflect this fact. Thus, rate in dinvgamma() refers to the rate parameter of dgamma(), the rate parameter of the gamma distribution.

This forms a kind of linguistic problem: the parameter rate of dgamma() is called rate because it is a rate parameter, which has a technical definition outside of the gamma/inverse gamma context. The inverse gamma distribution has a rate parameter in this other sense as well, but that rate parameter is not the same as the rate parameter of the gamma distribution. Worse: the rate parameter of the inverse gamma is the *inverse* of the rate parameter of the gamma distribution, which is the scale parameter. Thus, while (for example) dinvgamma() admits a parameter called rate, it is referring to the rate of the rate of the associated gamma distribution, not the rate parameter of the inverse gamma distribution, and this has naturally led to some confusion.

An analogy of the naming convention oddities may be helpful here to explain how these clashes occur. The lognormal distribution $LogN(\mu, \sigma^2)$ is, by definition, the distribution of a random variable whose log is a normal $N(\mu, \sigma^2)$ distribution, *i.e.*, $Y = e^X$ with $X \sim N(\mu, \sigma^2)$: a lognormal is the exponential of a normal. Throughout statistics $\mu$ conventionally denotes the mean of a random variable or distribution and $\sigma^2$ its variance. However, although the first parameter of the lognormal is conventionally denoted $\mu$, the mean of a lognormal random variable is not this $\mu$, and similarly $\sigma^2$ is not its variance. Users simply have to know that they are the mean and variance of the "distribution behind the distribution." This is a bit confusing and goes against the convention that $\mu$ ($\sigma^2$) is the mean (variance) of the distribution. It is a clash of conventions that the community deals with and accepts as a fact of life, albeit in some cases begrudgingly.

Base R implements the lognormal distribution as dlnorm() with formal arguments meanlog and sdlog to try to clarify this distinction, but this was not done when initially writing **invgamma**, and this has been a source of confusion. When writing **invgamma**, our original intention was to write other similar packages that do the same thing: provide tools for other common distributions not in R that are simple transformations of distributions that are. The convention was to maintain the parameters of the "distributions behind the distributions." This makes sense, but it does make this part messy.

Unfortunately it is not something that can be changed at this point, as the package has been in use for nearly a decade with several hundred thousand downloads. Similar to the base distribution of R, we have adopted a long-term mindset lending significant deference to software interface continuity, and thus have opted to keep the naming the way it is. To avoid mishaps, invgamma issues a package startup message alerting users to this issue. It is

possible that if a responsible way can be determined to gradually transition to a more transparent naming convention, the code base will shift, alerting users to the change.

## RELATED WORKS

As the inverse gamma distribution has important applications all over statistics, similar implementations of it have arisen in other user-contributed R packages. The two most prominent of these are the MCMCpack and actuar packages (*Martin, Quinn & Park, 2011*; *Dutang, Goulet & Pigeon, 2008*; *Dutang, 2016*). MCMCpack is designed for Markov chain Monte Carlo algorithms in Bayesian statistics, and actuar is designed for actuarial science in R. Both are large, domain-specific packages that are strange to list as dependencies in packages with applications in unrelated domains, *e.g.*, pharmaceutical statistics. Moreover, both of them have little quirks that make them less than ideal: MCMCpack only implements `dinvgamma()` and `rinvgamma()`, with restricted argument lists and a hard-coded version of the PDF; and actuar uses custom C-level implementations of the functions.

While not dramatically different than these, invgamma is preferable to these two packages as a general purpose implementation of the inverse gamma distribution for three reasons. First, invgamma is a small package, domain neutral, and has a name that reflects what it is, which is ideal for loading in other packages and for finding it. Second, invgamma leverages mathematical relationships between the gamma distribution and the inverse gamma distribution to allow the `*invgamma()` functions to wrap the `*gamma()` functions. Consequently, the `*invgamma()` functions are both efficient and automatically keep pace with changes to the gamma distribution functions made by the core team of R maintainers. Third, **invgamma** follows the same naming conventions and parameterizations as base R and has the same arguments as the `*gamma()` functions, so that using the `*invgamma()` functions are natural to R users. We note that actuar's implementation satisfies (3) as well. Additionally, invgamma includes `*invchisq()` and `*invexp()` functions for the closely related inverse $\chi^2$ (chi-squared) and exponential distributions.

## ACCURACY CONSIDERATIONS AND LIMITATIONS

One additional point that warrants a note is numerics concerning the transformations. **invgamma** was intended to be a lightweight, low-maintenance package implementing the inverse gamma, inverse chi-square, and inverse exponential distributions. It uses the transformation theorem in all cases. One of the challenges to using this implementation strategy is that the numerics are not optimized for the particular situation at hand. Arithmetic on a computer is not the same as arithmetic in theory, and as a consequence the best computer implementations of mathematical facts/algorithms are tailored to the specific cases at hand. Since these are not optimized in this way, questions may arise to the extent of the validity of the functions. The biggest questions surround the random number generators.

One way to test the values produced by the RNGs is *via* goodness-of-fit (GoF) testing of their draws. From an inferential perspective this is analogous to the graphics presented in Fig. 2, but operationally it is more mechanical as it does not require judgment calls, at least

not directly. The basic idea is to take a large number of draws $n$ from the presumed distribution and assess those with a GoF test where $H_0 : F_X = F_0$, where $F_X$ represents the distribution of the draws and $F_0$ represents the claimed distribution. If we conduct such a procedure at points ranging across the $(\alpha, \lambda)$ parameter space, we can get a good feel for which ranges of the parameter space the RNGs break down.

To code such a simulation, we begin with a function that performs such a GoF test, the Kolmogorov-Smirnov (KS) test, for a specific shape/rate combination. The following code accepts an $(\alpha, \lambda)$ shape/rate parameterization of the inverse gamma and uses the built-in function `ks.test()` to generate a $p$-value corresponding to the KS test for the inverse gamma distribution:

We then create a grid across a wide range of the parameter space, using a logarithmic

```r
test_rinvgamma <- function (shape, rate, n = 1e5) {

    sample <- rinvgamma (n, shape, rate)

    ks.test (sample, function (p) pinvgamma (p, shape, rate)) $ p.value

}

test_rinvgamma (shape = 3, rate = 7)

## [1] 0.4811224
```

spacing for better representation. We conduct the simulation at each of the points on the grid. Under the null hypothesis that the sampler is working correctly, at the 5% level roughly one out of every 20 tests will reject by design, so if we see regions that seem to have a higher rejection rate, those regions should be considered suspect. Figure 3 contains the results based off of the following code.

```r
# create the grid over the parameter space
n_grid <- 51
param_vals <- 10 **seq(-4, 4, length.out = n_grid)
param_grid <- expand_grid ("shape" = param_vals, "rate" = param_vals)
# run the simulations (in parallel)
install_if_needed("furrr" ); library ("furrr"); furrr_options (seed =
  TRUE)
## <furrr_options>
install_if_needed ("parallelly"); library ("parallelly")
plan(multisession(workers = min(availableCores(), 10)))
param_grid <- param_grid |>
  mutate ("p_val" = future_map2_dbl (shape, rate, test_rinvgamma))
# plot the results
```

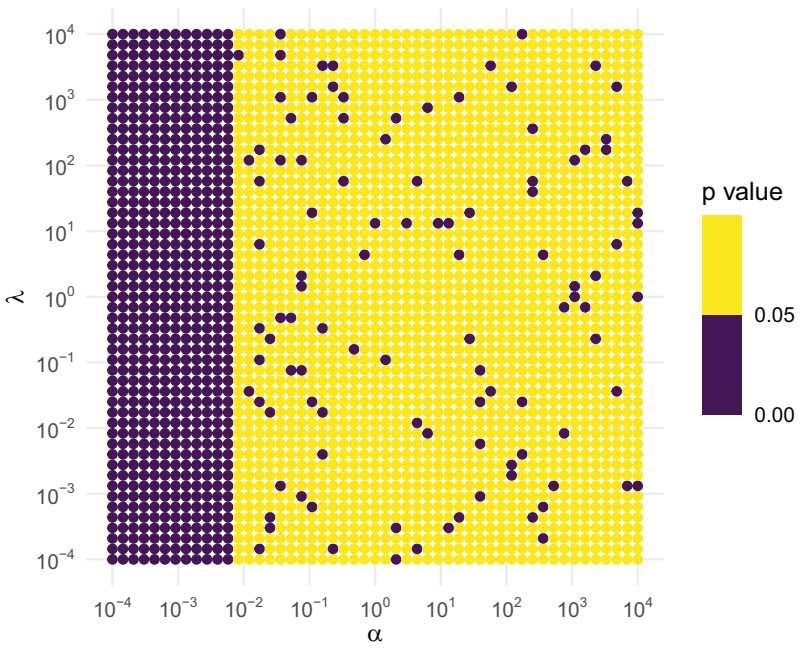

**Figure 3** KS GoF tests of inverse gamma draws for different parameter values.

```
fmt <- scales:: math_format (10**.x)

ggplot (param_grid, aes (shape, rate, color = p_val)) +
  geom_point () +
  scale_x_log10 (expression (alpha), n.breaks = 10, labels = fmt (-5:5)) +
  scale_y_log10 (expression (lambda), n.breaks = 10, labels = fmt(-5:5)) +
  scale_color_binned(breaks = c(0, .05, 1)) +
  labs(color = "p value") +
  coord_equal ()
```

If the RNG were correct, roughly 1 out of every 20 points should be purple. As this is violated for small shapes, the RNG is not to be trusted for small $\alpha$. Figure 3 clearly suggests that this is the case when $\alpha$ is less than about 0.01, regardless of what the value of $\lambda$ is. As a consequence, `rinvgamma()` issues a warning in these situations:

```
rinvgamma(4, shape = .001, rate = 3)

## Warning: `rinvgamma()` is unreliable for `shape` <=.01.

## [1]        Inf        Inf        Inf 1.528909e+113
```

The reason for this failure is quite evident: when the shape is low, the gamma distribution piles its mass near 0. When the draws are inverted, they are enormous, frequently overflowing the floating point number system to return infinity. This problem is inherited from R's RNG for the gamma distribution, for which the corresponding values are identically zero:

```
rgamma(10, shape = .001, rate = 3) == 0

## [1] FALSE FALSE FALSE FALSE FALSE FALSE TRUE FALSE TRUE TRUE
```

Similar R problems exist with duplications, even when the sampler is accurate:

```
rgamma (1e7, shape = 3, rate = 7) |>
table () |> sort(decreasing = TRUE) |> head (n = 5)
##
## 0.265165629569841 0.269360625139932 0.2955693295218 0.305498470042189
##                 2                 2                2                 2
## 0.61988692622084
##                2
```

Both of these illustrate that the problems that `rinvgamma()` has are limitations inherited from the R's `rgamma()` itself and are quite mild. Because the functions of invgamma are tied to these implementations, they will be corrected automatically when `rgamma()` is.

Similar tests can be conducted for the inverse chi-squared and inverse exponential distributions. The inverse chi-squared distribution is similar to the inverse gamma. Figure 4 illustrates that the inverse chi-squared RNG fails when $v \leq 0.01$ and $ncp \leq 10$ or so, and thus the RNG issues a warning in this setting.

```
test_rinvchisq <- function (df, ncp, n = 1e5) {
  sample <- rinvchisq(n, df, ncp)
  ks.test(sample, function(p) pinvchisq(p, df, ncp))$p.value
}
expand_grid("df" = param_vals, "ncp"= param_vals) |>
  mutate ("p_val" = future_map2_dbl (df, ncp, test_rinvchisq)) |>
  ggplot (aes(df, ncp, color = p_val)) +
    geom_point() +
    scale_x_log10(expression(nu), n.breaks = 10, labels = fmt(-5:5))+
    scale_y_log10("ncp", n.breaks = 10, labels = fmt(-5:5)) +
```

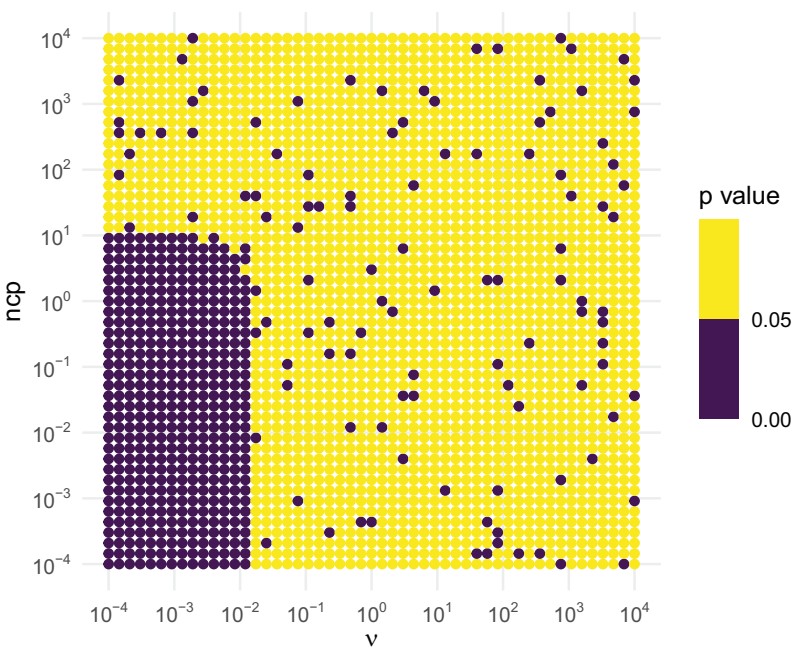

**Figure 4 KS GoF tests of inverse chi-squared draws for different parameter values.**

```
    scale_color_binned(breaks = c(0, .05, 1)) +

    labs(color = "p value") +

    coord_equal()
```

The inverse exponential is the easiest as it only admits a single parameter, `rate`. Unlike the others, the simulations illustrated in Fig. 5 suggest that the inverse exponential RNG is robust across a wide range of parameter values.

```
test_rinvexp <- function (rate, n = 1e5){

  sample <- rinvexp (n, rate = rate)

  ks.test (sample, function (p) pinvexp (p, rate))$p.value

}
tibble ("rate" = 10**seq(-4, 4, length.out = 2*n_grid)) —>

  mutate ("p_val" = future_map_dbl (rate, test_rinvexp)) —>

  ggplot (aes(rate, 0, color = p_val)) +

    geom_point () +

    scale_x_log10 (expression (lambda), n.breaks = 10, labels = fmt
    (-5:5)) +

    scale_color_binned(breaks = c(0, .05, 1), guide = FALSE)+
```

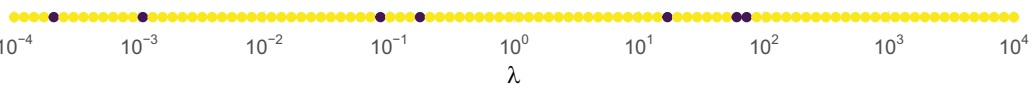

**Figure 5 KS GoF tests of inverse exponential draws for different parameter values.**

```
theme (axis.text.y = element_blank (), axis.title.y = element_blank
(), panel.grid.major.y = element_blank()) +
coord_equal ()
```

## CONCLUSIONS

In this article we have surveyed the invgamma R package, along with documenting the mathematics motivating the functions provided and tests of their accuracy. The package functions *invgamma(), *invchisq(), and *invexp() are largely self-maintained through their connections to the *gamma(), *chisq(), and *exp() functions provided in the base distribution of R and efficient for the same reason. The package is light weight and ideal for including in other R packages.

## ACKNOWLEDGEMENTS

The authors thank Keefe Murphy for private correspondence fixing errors in the quantile function with log.p = TRUE.

### Funding
The authors received no funding for this work.

### Competing Interests
The authors declare that they have no competing interests.

### Author Contributions
- David Kahle conceived and designed the experiments, performed the experiments, analyzed the data, performed the computation work, prepared figures and/or tables, authored or reviewed drafts of the article, and approved the final draft.
- James Stamey conceived and designed the experiments, analyzed the data, authored or reviewed drafts of the article, and approved the final draft.

### Data Availability
The code is available at GitHub and Zenodo:

- https://github.com/dkahle/invgamma.

- David Kahle. (2025). dkahle/invgamma: v1.2 (v1.2). Zenodo. https://doi.org/10.5281/zenodo.16944077.

## Supplemental Information

Supplemental information for this article can be found online at http://dx.doi.org/10.7717/peerj-cs.3205#supplemental-information.

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
