# Peer review of "invgamma: the inverse gamma distribution in R"

_PeerJ Computer Science, doi:10.7717/peerj-cs.3205_

## Round 0.1 · original submission · Minor Revisions

Thank you for submitting this manuscript to PeerJ Computer Science. The two reviews I've received are both very positive, and I'd be grateful if you could look at the reviewers' comments and address them or respond as appropriate.

·

Basic reporting

• Clear and unambiguous, professional English used throughout.
• Some literature references issues and recommended improvements:
o Although a link is provided to CRAN in the abstract, there is not one in the body of the paper. Additionally, it is this reviewer’s preference that URLs be shown in full rather than as link to text. Specifically, how authors have done links on page 2 for DLMF, R source code, etc. is preferred.
o Line 106: First reference to pgamma() function call doesn’t have the footnote for source of the function call. The footnote occurs on line 109.
o Line 248: This is the second use of this same reference in an almost identical context to line 41. Typically a reference need only be listed in body of the text the first time used.
o Some references appear to be missing details on what journal or on-line source the content can be found at. For example:
* “Muenchen, R. A. (2016). The popularity of data analysis software.” Is this a book, blog post, journal article?
* “Csa´rdi, G. (2019). cranlogs: Download Logs from the ’RStudio’ ’CRAN’ Mirror. R package version 341 2.1.1.” For open source software, a URL is expected – or how to install/acquire the software as these authors show for the “remotes” package. Additionally, running citation("cranlogs") shows a URL.
* “Dutang, C. (2016). CRAN Task View: Probability distributions.” Is this a journal article, website, something else? More details are needed.
* “Pedersen, T. L. (2024). patchwork: The Composer of Plots. R package version 1.3.0.” I recommend providing the URL for this package as seen in citation(“patchwork”)

• Professional article structure, figures, tables. Raw data shared.
o One recommended improvement: Make code in paper available as an .R, .Rmd, .qmd, .txt, or similar file. The code from the paper does not appear to be in the supplementary materials.
o Additionally, although the users are shown how to install invgamma, many other packages are required to run the code examples. A note or comment detailing all the additional required installations would be helpful.
• Self-contained with relevant results to hypotheses.
• Formal results include clear definitions of all terms and theorems, and detailed proofs.

Experimental design

No recommended changes based on this aspect.

Authors have met requirements:
• Original primary research within Aims and Scope of the journal.
• Research question well defined, relevant & meaningful. It is stated how this software package fills an identified knowledge gap.
• Rigorous investigation performed to a high technical & ethical standard.
• Methods described with sufficient detail & information to replicate. Package source code and example code provided in supplement, URL, and in some cases, body of the text.

Validity of the findings

No recommended changes based on this aspect.

Authors have met requirements:
• Source code and example code provided and verified to work by this reviewer. A few recommendations for improvement have already been mentioned.
• All underlying data have been provided; they are robust, statistically sound, & controlled.
• Conclusions are well stated, linked to original research question & limited to supporting results.

Additional comments

Line 37 “Conspicuously missing from the list, however, is the inverse gamma distribution and its related sub-families…” Why was inverse gamma not included? Is/was this feature deemed not important or useful to most users? Was there (is there still) some technical obstacle?

Line 39, 217, 276, 321 “…self-maintaining” / “self-maintained” I think this word choice is confusing. This makes me think of the term “self-healing” which is used in computing and computing marketing for some 20+ years. See this reference:

@article{Yazdanparast2024Mar,
author = {Yazdanparast, Zahra},
title = {{A Survey on Self-healing Software System}},
journal = {ArXiv e-prints},
year = {2024},
month = mar,
eprint = {2403.00455},
doi = {10.48550/arXiv.2403.00455}
}

The invgamma gamma package has no dependencies outside base R, thus should be easy to maintain, so perhaps more explanation is needed or use alternative terms such as “low-maintenance ” or “no external dependencies.”

Line 40, 247: “with downloads totaling in the hundreds of thousands” Using the cranlogs package I see > 300,000 downloads as of now. I think “in the hundreds of thousands” is more vague and could undersell the importance and breadth of adoption of this software package.

r$> library(cranlogs)
cd_output <- cran_downloads(package="invgamma", from="2015-01-01", to="2025-12-31")
sum(cd_output$"count")
Warning message:
In fill_in_dates(df, as.Date(res1$start), as.Date(res1$end)) :
Time interval in the future
[1] 337604

First downloads on 2016-06-24

Line 247 “Unfortunately it is not something that can be changed at this point, as the package has been in use for nearly a decade with several hundred thousand downloads”
I think this deserves more discussion. I agree with the authors perspective and think this is an important consideration in Computer Science. Some readers and users of invgamma may not understand the importance of software interface continuity. Additionally, this long-term view is often not followed by many popular R packages that change function parameters at minor releases.

Line 274: “A COMMENT ON NUMERICS”
This is interesting and likely of interest to readers. However, at a length of 4 pages with multiple code blocks and 3 figures, it seems more substantial than a “comment.”

Code block after line 298:
• Code block has “10^seq” and “10^.x”– but in the PDF rendering process the regular caret (ASCII character 94) becomes a superscript caret which is a non-keyboard symbol (ASCII character 136) so cannot be run by my R session.
• “plan(multisession(workers = parallelly::availableCores()))” This code fails on Linux systems with more than 128 cores. "Error: Cannot create XXX parallel PSOCK nodes. Each node needs one connection, but there are only 124 connections left out of the maximum 128 available on this R installation. To increase this limit in R (>= 4.4.0), use command-line option '--max-connections=N' when launching R." I don’t think even 124 workers is needed for this code. Either a comment or upper bound check would be beneficial.

Cite this review as

·

Basic reporting

The manuscript is self-contained, and the package is sufficiently light weight making it easy to integrate in other R packages. However, a subtle improvement would be to explicitly state the package limitations under a dedicated subsection such as "Limitations" for better visibility. For instance, there is a limitation on edge cases inherited from rgamma(). The package notes this issue via a runtime warning e.g. the numerical overflow when shape is <= 0.01, however in the manuscript, this is buried in the "A Comment on Numerics" section (lines 299-314). While the warning itself is appropriate, users browsing the manuscript might miss this constraint. I suggest adding a "Limitations" subsection under the "A Comment on Numerics" to consolidate the edge cases and loss of precision for extreme values. Notably, figures 3 to 5 empirically validate limitations. Overall, the basic reporting is excellent.

Experimental design

The package startup message alerting users to the parameterisation issue (lines 216:251) is an excellent safeguard. The theoretical explanation of the relationship between rate/scale is sound. To further reduce potential confusion, consider adding a commented practical example in the documentation as is in base R's dgamma(). For example, showing the equivalence of the parameterisation dinvgamma(x, shape = 2, rate = 3) and dinvgamma(x, shape = 2, scale = 1/3).

Validity of the findings

no comment

Additional comments

While the package's single-thread design is ideal to maintain the lightweight nature, users working with very large samples (e.g., n=1e7) may experience performance challenges. I suggest adding a parallel computing example e.g.in the README (without modifying the main package code). This will keep the package lightweight while guiding power users.

Cite this review as

---

## Round 0.2 · accepted · Accept

Taking into account the reviewers’ comments, I'm pleased to accept this manuscript for publication in PeerJ Computer Science. Thank you for this contribution to the journal!

·

Basic reporting

Looks great - all previously identified issues addressed. No additional issues to report.

Experimental design

Looks great.

Validity of the findings

Looks great.

Additional comments

The diff file didn't have all changes made between previous review and this review. However, from a combination of reviewing the diff file and comparison of the new version with the old version, all my comments were addressed.

The new .R file is a great addition to the code provided in the article.

Cite this review as